# Rose Virome Analysis and Identification of a Novel Ilarvirus in Taiwan

**DOI:** 10.3390/v14112537

**Published:** 2022-11-16

**Authors:** Tsung-Chi Chen, Yu-Chieh Lin, Chian-Chi Lin, Yi-Xian Lin, Yuh-Kun Chen

**Affiliations:** 1Department of Medical Laboratory Science and Biotechnology, Asia University, Wufeng District, Taichung 41354, Taiwan; 2Department of Plant Pathology, National Chung Hsing University, South District, Taichung 40227, Taiwan

**Keywords:** diagnostics, rose virus, *Capillovirus*, *Ilarvirus*, *Luteovirus*, *Partitiviridae*, RNA-seq

## Abstract

Rose (*Rosa* spp.), especially *R. hybrida*, is one of the most popular ornamental plants in the world and the third largest cut flower crop in Taiwan. Rose mosaic disease (RMD), showing mosaic, line patterns and ringspots on leaves, is a common rose disease caused by the complex infection of various viruses. Due to pests and diseases, the rose planting area in Taiwan has been decreasing since 2008; however, no rose virus disease has been reported in the past five decades. In the spring of 2020, rose samples showing RMD-like symptoms were observed at an organic farm in Chiayi, central Taiwan. The virome in the farm was analyzed by RNA-seq. Rose genomic sequences were filtered from the obtained reads. The remaining reads were de novo assembled to generate 294 contigs, 50 of which were annotated as viral sequences corresponding to 10 viruses. Through reverse transcription-polymerase chain reaction validation, a total of seven viruses were detected, including six known rose viruses, namely apple mosaic virus, prunus necrotic ringspot virus, rose partitivirus, apple stem grooving virus, rose spring dwarf-associated virus and rose cryptic virus 1, and a novel ilarvirus. After completing the whole genome sequencing and sequence analysis, the unknown ilarvirus was demonstrated as a putative new species, tentatively named rose ilarvirus 2. This is the first report of the rose virus disease in Taiwan.

## 1. Introduction

Rose (*Rosa* spp.), especially *R. hybrida*, is one of the world’s most popular and important ornamental plants. Due to their elegant and fragrant flowers, rose is widely used in landscaping, cut and potted flowers, food ingredients, cosmetics and for extracting essential oils. The estimated annual production for landscaping was around 15–18 billion stems and 220 million roses in 2003 [1,2]. The global market value for the rose was 24 billion Euros from 1995 to 2007 [3]. Total cut rose transaction value in 2020 reached $198 million [4]. From 2012 to 2013, the global cut rose planting area was 60,447 ha, and the top ten cut rose-producing countries in the world were India, China, Ecuador, Colombia, Kenya, Mexico, Italy, Thailand, Japan and the Netherlands. The Netherlands was the largest exporter of cut rose, accounting for 42.32% of total world exports of cut rose [5]. The entire rose industry on a worldwide scale has an economic impact of tens of billions of dollars [6].

A wide range of pathogens, including fungi, bacteria, viruses, nematodes and phytoplasmas, drastically affect the ornamental value of cultivated roses around the world by leaf and flower mosaics, distortion, spotting, discoloration, necrosis, abscission, reduced growth and plant death [6]. The viral disease is considered one of the biggest threats to rose cultivation due to the difficulty of control. Virus infection may cause inconspicuous symptoms in rose plants that are often overlooked. Therefore, viruses can be easily spread by vegetative propagation during budding and grafting procedures [7]. Rose rosette disease (RRD) and rose mosaic disease (RMD) are the most important rose virus diseases. RRD commonly exhibits a witches broom symptom with unusual thorniness and reddening of shoots, distorted flowers, stunting, defoliation and eventual plant death that causes significant damage to landscape rose. Rose rosette virus (RRV) of the genus *Emaravirus* is the causal agent of RRD and can be effectively spread by the eriophyid mite (*Phyllocoptes fructiphilus*) [6,8]. RMD is generally associated with mixed infections of viruses that belong to different taxa, especially apple mosaic virus (ApMV) and prunus necrotic ringspot virus (PNRSV) of the genus *Ilarvirus* and Arabis mosaic virus (ArMV) and strawberry latent ringspot virus (SLRSV) of the genus *Nepovirus* [9]. Symptoms associated with RMD are highly variable, depending primarily on rose cultivars, virus agents and environmental factors. Some of the common foliar symptoms include chlorotic line patterns, ring spots, yellow vein banding and puckering, intense yellow spots, mottling and severe distortion [10]. In addition, flower distortion, reduced flower production and flower size, stem caliper at the graft union and reduction in vigor, early autumn leaf drop, lower bush survival rates, increased susceptibility to cold injury and more difficult establishment after transplanting are shown. RMD is thought to have been propagated in roses by grafting from infected rootstocks or scions, subsequently spreading among rose cultivars [11].

Serological and molecular methods are commonly deployed for virus detection and diagnosis. Enzyme-linked immunosorbent assay (ELISA) is a widely used serological technique for routine virus testing in phytosanitary, quarantine and virus certification. Polymerase chain reaction (PCR), commonly reverse transcription (RT)-PCR used for RNA viruses, is a relatively sensitive method for virus detection. The advanced real-time or quantitative PCR (qPCR) adds further benefits in a routine testing laboratory primarily due to its higher sensitivity and speed, with no need to use agarose gel electrophoresis. Loop-mediated isothermal amplification (LAMP) is suitable for on-site use [12]. The microarray can be applied to simultaneously detect and differentiate closely related viruses [13]. Recently, the clustered regularly interspaced short palindromic repeats (CRISPR) system, widely applied for genome editing, is also applied in virus diagnosis [14]. Although these methods are highly specific and sensitive, they are not suitable for comprehensive surveys of viruses in regional crops. Next-generation sequencing technologies, also known as high-throughput sequencing (HTS), can sequence millions or billions of DNA molecules in parallel and have become a powerful tool for metagenomics research [15]. HTS can also perform virome studies to detect all known and unknown viruses present in a plant [16]. Despite HTS allowing the elucidation of the elusive etiology of some diseases, it is not sufficient to find direct associations between the disease and viruses detected in the infected plant. Therefore, the validation of viruses with other techniques, such as Koch’s postulates and PCR, is still necessary [17].

Through HTS, many rose viruses have been discovered in recent years. Thus far, at least 26 viruses have been reported infecting roses worldwide. They include five *Ilarvirus* spp., ApMV, PNRSV, tobacco streak virus (TSV) [18], blackberry chlorotic ringspot virus (BCRV) [19,20] and rose ilarvirus-1 (RIV-1) [21]; four *Nepovirus* spp., ArMV, SLRSV, tobacco ringspot virus and tomato ringspot virus [18]; four *Orthotospovirus* spp., impatiens necrotic spot virus (INSV), iris yellow spot virus (IYSV), tomato spotted wilt virus (TSWV) and tomato yellow fruit ring virus (TYFRV) [22,23]; three *Carlavirus* spp., rose virus A (RVA) [24], rose virus B (RVB) [25] and rose virus C (RVC) [26]; two species of the family *Partitiviridae*, rose cryptic virus 1 (RoCV-1) [27,28] and rose partitivirus (RoPV) [29]; one *Capillovirus* sp., apple stem grooving virus (ASGV) [30]; one *Emaravirus* sp., RRV [8]; one *Luteovirus* sp., rose spring dwarf-associated virus (RSDaV) [31]; one *Roymovirus* sp., rose yellow mosaic virus [32]; one *Closterovirus* sp., rose leaf rosette-associated virus (RLRaV) [30]; one *Cytorhabdovirus* sp., rose virus R (RVR) [33]; one *Begomovirus* sp., rose leaf curl virus (RoLCuV) [34]; and one *Rosadnavirus* sp., rose yellow vein virus (RYVV) [35]. Mixed infections between these viruses are frequently detected.

Rose is the third largest cut flower crop in Taiwan with an annual output value of $20 million, and Nantou is the largest rose planting area with 106.7 ha, followed by Taichung (30.0 ha), Pingtung (22.0 ha) and Changhua (14.4 ha) [36]. Since 2008, the acreage of rose plantations in Taiwan has decreased because of serious pests and diseases [37]. Fungal diseases are common on rose plants and include anthracnose (*Colletotrichum gloeosporioides*), gray mold (*Botrytis cinerea*), black spot (*Diplocarpon rosae*), black rot (*Calonectria cylindrospora*), powdery mildew (*Erysiphe simulans* var. *simulans*) and stem canker (*Paraconiothyrium fuckelii*) [38]. However, there have been no reports of rose virus disease in the past five decades. In the spring of 2020, rose plants showing RMD-like symptoms were noticed at an organic rose farm in Chiayi, central Taiwan. HTS was performed for disease diagnosis and RT-PCR methods were developed for virus detection. Finally, a total of seven viruses, including six known rose viruses, ApMV, ASGV, PNRSV, RoPV, RSDaV and RoCV-1, and a novel ilarvirus were detected. The whole genome sequence of the newly discovered ilarvirus was further determined, indicating that this virus is a putative new *Ilarvirus* species, named rose ilarvirus 2 (RIV-2). The occurrence of these rose viruses was also addressed in this study.

## 2. Materials and Methods

### 2.1. Sample Collection

Rose leaves with virus-like symptoms were sampled from the organic rose farm in Fanlu Township, Chiayi County and the campus of Asia University in Wufeng District, Taichung City in central Taiwan in 2020–2022 for virus identification and detection. Rose seedlings were also collected from nurseries in Nantou and Changhua in central Taiwan in 2021–2022 for virus testing. All samples were stored at −80 °C for long-term storage.

### 2.2. HTS and Bioinformatics Analysis

Foliar total RNA of symptomatic rose samples was extracted using the Total RNA Miniprep Purification kit (GMbiolab, Taichung, Taiwan) following the manufacturer’s instructions. DNA was removed from the total RNA extract using the Invitrogen™ TURBO DNA-*free*™ kit (Thermo Fisher, Waltham, MA, CA, USA). The cDNA library was constructed using the Universal RNA-Seq Library Preparation kit with NuQuant (Tecan, Männedorf, Switzerland). The quantity and quality of the cDNA library were analyzed by Qubit^®^ 2.0 Fluorometer with the dsDNA High Sensitivity kit (Thermo Fisher, Waltham, MA, USA) and Agilent 2100 Bioanalyzer (Agilent, Santa Clara, CA, USA). Paired-end sequencing was performed by Tri-I Biotech Inc. (New Taipei City, Taiwan) using the NovaSeq 6000 Sequencing System (Illumina, San Diego, CA, USA). Paired-end reads are 151 bp in length. Raw data quality was checked and trimmed with CLC Genomics Workbench v10 [39]. The trimmed paired reads were mapped and analyzed by CLC Genomics Workbench v10, and the reads were pasted back to the reference sequences in the database of the National Center for Biotechnology Information (NCBI) (https://www.ncbi.nlm.nih.gov/genome/?term=Rosa+chinensis, accessed on 15 January 2019) to remove rose genome sequences and kept the unmapped reads. The unmapped reads were de novo assembled to generate contigs with SPAdes v3.13.0 [40]. The contigs were analyzed by the Basic Local Alignment Search Tool (BLAST) of NCBI (https://blast.ncbi.nlm.nih.gov/Blast.cgi) to identify viruses. BLASTn/BLASTx analyses of the contigs were performed against local and online databases.

### 2.3. Virus Detection

Primer pairs (Appendix A) designed from virus-annotated contigs were used to verify the presence of viruses in the field rose samples. The One-step RT-PCR Kit III (GMbiolab, Taichung, Taiwan) was used for virus detection. RT-PCR was performed using 1 μg of total RNA as the template and 200 nM of each primer at the final concentration. Aliquots of the 25 μL reaction mixture also consisted of 1× Reaction mix (200 μM dNTPs, 1.5 mM Mg^2+^ and enzyme stabilizer), 1× Enhancer, enzyme mix of reverse transcriptase (RTase) and *Taq* DNA polymerase, and RNase block according to the manufacturer’s recommendation. The RT-PCR conditions were set as 30 min for cDNA synthesis at 50 °C, and 2 min for RTase inactivation at 94 °C, followed by 35 cycles of 30 s for denaturation at 94 °C, 30 s for annealing at 56 °C, and 1–1.5 min (depending on the product size) for synthesis at 72 °C and a final cycle of 72 °C for 7 min. The amplified DNA fragments were analyzed by electrophoresis in 1% agarose gels and then eluted from gels using the Micro-Elute DNA Clean/Extraction kit (GMbiolab, Taichung, Taiwan) following the instruction of the manufacturer. The eluted PCR products were ligated to the pCR2.1-TOPO vector using the Invitrogen™ TOPO TA Cloning kit (Thermo Fisher, Waltham, MA, USA) and the recombinant plasmids were transferred into *E. coli* DH5α competent cells by Invitrogen™ One Shot™ transformation reaction (Thermo Fisher, Waltham, MA, USA). The plasmid DNAs of selected clones were isolated using the NautiaZ Plasmid DNA Extraction Mini kit (NAUTIA GENE, Taipei, Taiwan), three clones per PCR product. Nucleotide (nt) sequencing was carried out by Mission Biotech Co., Ltd. (Taipei, Taiwan) using the Applied Biosystems™ 3730XL automated DNA sequencing system (Thermo Fisher, Waltham, MA, USA).

### 2.4. Sanger Sequencing for Genome Sequencing of RIV-2

Total RNA extracted from RIV-2-positive rose leaves was used as the template. Two-step RT-PCR was performed for amplification. Mixtures consisting of 2 μg of total RNA mixed with 200 nM of individual reverse primers designed from the annotated contig sequences (Appendix A) and 200 U Invitrogen™ SuperScript™ IV (SSIV) RTase (Thermo Fisher, Waltham, MA, USA) were incubated at 50 °C for 10 min to synthesize cDNAs, and then the reaction was inactivated by heating at 80 °C for 10 min. Subsequently, 2 μL of cDNAs were mixed with Blend Taq™-Plus reagent (Toyobo, Osaka, Japan) for a total volume of 25 μL, consisting of Blend Taq™-Plus *Taq* DNA polymerase (1.25 U), dNTPs (200 μM), forward and reverse primers (200 nM each) (Appendix A) and 1× PCR buffer, and heated at 94 °C for 3 min for the hot start. PCR was performed by 35 cycles of strand separation at 94 °C for 30 s, annealing at 58 °C for 1 min, and synthesis at 72 °C for 3–3.5 min depending on the product size. A final reaction was conducted at 72 °C for 7 min. The amplified DNA fragments were cloned by the Invitrogen™ TOPO TA Cloning kit (Thermo Fisher, Waltham, MA, USA) for sequencing as aforementioned. Three clones per amplicon were sequenced.

Rapid amplification of cDNA ends (RACE) was performed to verify both 5′ and 3′ ends of the genome of RIV-2, as described previously [41]. Total RNA was denatured by heating at 70 °C for 10 min and then put on ice for 1 min. The first strand cDNAs were synthesized by SSIV RTase (200 U) mixed with 200 nM of proper primers (Appendix A) at 50 °C for 60 min, and then the reaction was stopped at 70 °C for 15 min. After the removal of template RNA from RNA-cDNA hybrids by RNaseH (New England Biolabs, Ipswich, MA, USA), the cDNA molecules were precipitated by adding 1/10 volume of 3 M sodium acetate (pH 5.2) and 2.5 volume of absolute ethanol at −80 °C for 2 h. After centrifugation at 14,000 rpm for 15 min, the pellets were resuspended in 20 μL of RNase-free water. Subsequently, cDNA fragments were tailed with 200 nM PolyG (11g3a3g) (5′-GGGGGGGGGGGAAAGGG-3′) adapter [41] by 20 U terminal deoxynucleotidyl transferase (TdT) (New England Biolabs, Ipswich, MA, USA). The tailed cDNA fragments were used as templates for PCR amplification by mixing with 200 nM PolyC (3c3t1lc) (5′-CCCTTTCCCCCCCCCCCCC-3′) primer [41] complementing to the polyG tail, 200 nM proper primers (Appendix A) and Blend Taq™-Plus reagent (Toyobo, Osaka, Japan) mentioned above. PCR was performed by 35 cycles of strand separation at 94 °C for 30 s, annealing at 58 °C for 1 min, and synthesis at 72 °C for 1 min. The amplified DNA fragments were cloned by the Invitrogen™ TOPO TA Cloning kit (Thermo Fisher, Waltham, MA, USA) for sequencing, as aforementioned, three clones per amplicon.

### 2.5. Sequence Analysis of RIV-2 Genome

The complete genome sequences of 24 ilarviruses available in the GenBank databases (Appendix A) were used for analysis. The nt and amino acid (aa) sequence identities of the full-length genome and open reading frames (ORFs) of the viruses were calculated using AlignX in the software Invitrogen™ Vector NTI Advance 10 (Thermo Fisher, Waltham, MA, USA). Multiple sequence alignments were performed by the ClustalX 2.1 program. Phylogenetic analyses were conducted using the Tree Explorer program of MEGAX [42], with 1000 bootstrap replicates. Phylogenetic branches were set as the Neighbor-Joining method.

## 3. Results

### 3.1. Symptom Observation

In a field survey conducted at an organic rose farm in Chiayi in 2020–2021, rose plants were observed to exhibit RMD-like symptoms, such as mosaic, mottling and yellow spots and flower color-breaking (Figure 1; Appendix A). The symptoms were also observed on landscape roses on the campus of Asia University in Taichung in 2021.

### 3.2. Virus Detection by HTS and Computing

Twelve rose samples collected from the rose farm in Chiayi in February 2021 were used for virus detection. Combining the total RNAs of the rose samples to construct a cDNA library for HTS, a total of 42,920,486 reads were obtained. After trimming, 42,862,516 reads were obtained, which were then used to remove rose genome sequences by mapping reference sequences in the NCBI database. The remaining 4,759,744 unmapped reads were used for de novo assembly to generate 294 contigs, with an N_50_ of 834 bp. The contigs were used for virus identification by BLAST analysis. Finally, 50 contigs were annotated as viral sequences, of which six contigs mapped to ApMV, four contigs for tomato necrotic streak virus (TomNSV, *Ilarvirus* genus), six contigs for PNRSV, three contigs for RoCV-1, two contigs for RoPV, three contigs for ASGV, two contigs for RSDaV, 20 contigs for RLRaV, three contigs for RVB and one contig for Plasmopara viticola lesion associated ourmia-like virus 33 (PvLaOV-33). The results of the reads analysis are summarized in Table 1 and Appendix A. Unlike most contigs that shared higher than 90% nt identity with the corresponding reference sequences, the contigs annotated as TomNSV and PvLaOV-33 but showing less than 90% identity were tentatively named rose ilarvirus 2 (RIV-2) and OurmiaX, respectively.

### 3.3. Verification of Virus Presence by RT-PCR

Primers (Appendix A) designed from the virus-annotated contigs were used to verify the presence of the viruses in collected rose samples by RT-PCR. At least two amplicons of each virus were sequenced and analyzed to confirm viral identity. A total of seven viruses were detected, and the detection results are shown in Figure 2. Using the primer pairs AMV-R3-137-F/AMV-R3-1161-R and AMV-R3-940-F/AMV-R3-1811-R, two expected fragments of 1025 bp and 872 bp, containing the full-length ORFs of movement protein (MP) and coat protein (CP) of ApMV, respectively, were amplified. The sequences of the fragments were assembled to obtain a near-complete RNA3 sequence, which is 1674 bp long and has 98.2% identity to ApMV isolated in Yunnan, China (acc. no. AM490197.2), including 98.6% aa identity in MP and 99.1% aa identity in CP. The same strategy was also used to detect PNRSV and RIV-2. The primer pairs PNRSV-MPF1/PNRSV-MPR1 [43] and PNRSV-CPF1/PNRSV-CPR1 [44] were used to amplify the complete MP and CP ORFs of PNRSV with amplicon sizes of 983 bp and 699 bp, respectively. The sequences of the two fragments were merged to obtain a 1648-bp sequence that shared nearly 99% nt identity with PNRSV RNA3 such as acc. no. MN656194 from China. A 1788-bp near-complete RNA3 sequence of RIV-2 was obtained using two pairs of primers, RIV2-R3-1F/RIV2-R3-1R and RIV2-R3-2F/RIV2-R3-2R. The RIV-2 sequence shares the highest 78.8% identity with TomNSV RNA3 (acc. no. NC_039076). ASGV was detected by amplifying a 695-bp fragment using the primer pair ASGV140-F/ASGV834-R corresponding to ORF1. The amplicon sequence had the highest 91.1% nt identity to the ASGV L2 and T47 isolates (acc. no. MK599421 and KF434636) from China. RSDaV was detected by amplifying a 439-bp fragment using the primer pair RSDaV-F/RSDaV-R targeting the CP gene. The amplicon is up to 98.1% identical to the RSDaV genome sequence isolated in the USA (acc. no. EU024678). RoCV-1 was detected using the primer pairs RoCV1-R1-252-F/RoCV1-R1-808-R and RoCV1-R3-F/RoCV1-R3-R corresponding to dsRNA1 and dsRNA3, respectively. Both amplified dsRNA1 and dsRNA3 fragments, 557 bp and 762 bp, respectively, show 99.6–100% identity to those of RoCV-1 isolated in the UK (acc. no. MK075826 and MK075828), USA (acc. no. EU024675 and EU024676) and Canada (acc. no. KM598758 and KM598760). RoPV was detected by amplifying a 614-bp fragment using the primer pair RoParti-R1-956-F/RoParti-R1-1569-R targeting dsRNA1. The amplicon shares 98.9% identity with the dsRNA1 of RoPV isolated in Canada (acc. no. KU896858). RVB, RLRaV and the PvLaOV-33-related OurmiaX were not detected in all tested samples. In conclusion, three ilarviruses, ApMV, PNRSV and RIV-2, two partitiviruses, RoCV-1 and RoPV, one luteovirus, RSDaV, and one capillovirus, ASGV, have been validated to infect roses in Taiwan.

### 3.4. Molecular Characterization of RIV-2

One rose sample only positive for RIV-2 was used for whole-genome sequencing. Primers (Appendix A) designed from the annotated contigs were used for RT-PCR amplification. The RNA1 and RNA2 sequences were obtained from three overlapping fragments and the RNA3 sequence from four overlapping fragments. The 5′- and 3′-end sequences of RNA1, RNA2 and RNA3 were verified by RACE (Figure 3). The complete nt sequence of RNA1 was determined as 3408 nt in length that contains an ORF from nt 73 to nt 3213 for coding a 1046-aa (117.9 kDa) P1 protein. The RNA2 segment was determined as 3022 nt in length, containing two ORFs, P2 and 2b. The P2 ORF starts from nt 73 and ends at nt 2553 for coding an 826-aa (94.7 kDa) protein. The 2b ORF starts from nt 2144 and ends at nt 2833 for coding a 229-aa (26 kDa) protein. The RNA3 segment was determined as 2377 nt in length with two ORFs, MP and CP. The MP ORF starts from nt 406 and ends at nt 1308 for coding a 300-aa (33.2 kDa) protein. The CP ORF positions between nt 1327 and nt 2067 for coding a 246-aa (27.5 kDa) protein. The three genomic RNA segments possess consensus sequences ‘GGTATT’ at the 5′-end, and RNA1 and RNA2 have the same 5′-untranslatable region (UTR) sequence. In addition, the three genomic RNA segments share conserved sequences at the 3′-UTR. The genome organization of RIV-2 is illustrated in Figure 3. The genome sequences of RIV-2 have been deposited in the GenBank database, acc. no. ON843765-7.

The genome sequences of RIV-2 were compared with the available genome sequences of *Ilarvirus* species in the GenBank database. The analysis results are shown in Table 2. The three genomic RNA segments of RIV-2 shared 37.1–81.9% nt identity with the analyzed ilarviruses, the highest nt identity of 77.3–81.9% to TomNSV. Compared with other ilarviruses, RIV-2 has 50.7–84.3% nt identity (31.2–89.2% aa identity) for P1 ORF, 39.0–80.2% nt identity (27.7–83.4% aa identity) for P2 ORF, 44.3–78.1% nt identity (16.1–74.2% aa identity) for 2b ORF, 44.2–79.8% nt identity (14.0–81.4% aa identity) for MP ORF, and 40.3–78.6% nt identity (15.1–80.7% aa identity) for CP ORF. Phylogenetic analyses based on the aa sequence alignments of the P1, P2, 2b, MP and CP of 24 ilarviruses revealed that RIV-2 is clustered with members of subgroup 2, and is closely related to TomNSV and Tulare apple mosaic virus (TAMV), but also with the unclassified ilarvirus RIV-1 (Figure 4). Based on sequence similarity and phylogenetic relationship, RIV-2 was identified as a novel member of the genus *Ilarvirus*.

### 3.5. Filed Survey for Rose Viruses

During 2020–2022, 92 symptomatic rose samples were tested using the RT-PCR methods described above, of which 82 were from the disease outbreak farm in Chiayi and 10 were from the campus of Asia University in Taichung. The seven viruses, PNRSV, RIV-2, ApMV, RSDaV, RoCV-1, RoPV and ASGV, were detected in Chiayi with detection rates of 19.5%, 13.4%, 7.3%, 7.3%, 3.7%, 3.7% and 2.4%, respectively. In addition, three viruses, RoCV-1, RoPV and RSDaV, were detected in Taichung with detection rates of 40.0%, 20.0% and 10.0%, respectively (Table 3). On the other hand, 15 rose seedlings were collected for testing from two nurseries in Changhua in 2021 and one in Nantou in 2022. None of the above viruses were detected (data not shown).

## 4. Discussion

RMD is a well-known rose virus disease, but it may be overlooked by farmers due to symptoms similar to nutrient deficiencies. In the spring of 2020, the rose plants in the Chiayi organic rose farm suffered severe decline and death, causing the owner’s concern. To our knowledge, as mentioned above, at least 26 viruses have been reported to infect rose plants, making virus detection inconvenient. Therefore, the virome of the farm was analyzed using HTS technology in this study and, surprisingly, 10 viruses were annotated (Table 1). Of these, eight viruses have been reported to infect rose plants, namely ApMV, PNRSV, RoCV-1, RoPV, ASGV, RSDaV, RVB and RLRaV. Subsequently, the presence of ApMV, PNRSV, RIV-2, RoCV-1, RoPV, ASGV and RSDaV was confirmed by RT-PCR assay (Figure 2), and the correctness of the annotation was verified by amplicon sequencing. Although RVB, RLRaV and the PvLaOV-33-related OurmiaX were also annotated, they were not detected in the tested rose samples. However, their presence in Taiwan cannot be ruled out. The low reads count of 20 for OurmiaX, 46–67 for RVB and 20-1912 for RLRaV may indicate a lower incidence of these viruses.

RIV-2 was noted in the virome analysis because it has the closest but low nt identity (<90%) to the three genomic RNA segments of TomNSV, a member of the genus *Ilarvirus* in the family *Bromoviridae*. Based on the genomic characteristics, RIV-2 was further confirmed to be an ilarvirus, including: (1) the critical conserved motifs (S_88_-H_262_) of the methyltransferase (MET) domain and the conserved motifs (D_764_-K_1008_) of the helicase (HEL) domain in the P1 replicase [45], (2) the conserved motifs I-VIII (M_361_-R_605_) of the RNA-dependent RNA polymerases (RdRps) of positive-sense RNA viruses in P2 [46] and (3) a 2b ORF that overlaps the 3′ end of the P2 ORF [47]. According to the document of the International Committee on Taxonomy of Viruses (ICTV), serology, host range and sequence similarity are used for the demarcation of species within the *Ilarvirus* genus. However, the threshold for sequence similarity has not yet been defined [47]. In addition, based on molecular relationships, ilarviruses are clustered into four subgroups, of which subgroups 1 and 2 encode an additional 2b protein that functions as a viral suppressor of RNA silencing (VSR) [47,48]. The RIV-2 genome contains a putative 2b ORF and is phylogenetically related but distinct from TomNSV, TAMV and the newly discovered RIV-1 (Figure 4), indicating that RIV-2 is a novel species belonging to subgroup 2.

All three genomic RNAs of RIV-2 begin with a consensus ‘GGTATT’ sequence, similar to the previously reported GTATT sequence found in other species of the genus *Ilarvirus* [45]. The element may be crucial for the formation of specific structures that can be recognized by the ribosome and/or translation factors to initiate translation. In addition, the ‘GGTATT’ sequence can also be found at −5 to −10 upstream of the 2b ORF, implying that an internal ribosomal entry site (IRES) may be formed to facilitate the VSR expression. Furthermore, the 5′- and 3′-UTRs of RNA3 are significantly longer than those of RNA1 and RNA2 in RIV-2 as well as in other ilarviruses, and the 5′-UTR sequences of RNA1 and RNA2 of RIV-2 are identical, but different from that of RNA3 (Figure 3). They imply distinct regulatory mechanisms between RNA3 and RNA1/2, which encode the replicases P1/P2 and the VSR 2b to establish infection. The 3′-terminal sequences (23 nt) of the three genomic RNA segments of RIV-2 are almost identical (Figure 3). High conservation of UTRs among subgroup-1 ilarviruses has been proposed [45]; whereas, UTR conservation of the subgroup-2 ilarviruses could be addressed in the future. The 3′-UTRs of viral RNAs interacting with CP to initiate replication and establish infection in alfamoviruses and ilarviruses, termed ‘genome activation’, has been previously elucidated [49,50,51]. The conformational switch of the 3′-UTR from a tRNA-like structure (TLS) to a linear stem-loop structure (SLS) can be mediated by CP binding. The TLS favors viral replicase binding to synthesize viral antigenome RNAs. In contrast, CP binding triggers SLS formation to allow genomic RNA synthesis and translation [52,53]. Phylogenetic analysis suggests that all known ilarviruses might adopt the TLS [54].

The occurrence of the viruses was investigated for three years from June 2020 to August 2022, at the Chiayi organic rose farm in which the outbreak occurred and two years from July 2021 to September 2022 at the Taichung Asia University campus. Our results showed that ApMV, PNRSV, RIV-2, RoPV, RoCV-1, ASGV and RSDaV were detected in Chiayi. Surprisingly, mixed infections of up to five viruses could be detected in the organic farm in February 2021 (Appendix A). A large number of plants died, causing severe losses on this farm. The synergy between these viruses is worth paying attention to. The incidence of all viruses on the farm has dropped significantly since January 2022 due to the renewal of rose plants in the summer of 2021 (Table 3). The investigation is still ongoing. On the other hand, RoPV, RoCV-1 and RSDaV were also detected in Taichung. It suggests that RoPV, RoCV-1 and RSDaV may have been disseminated in central Taiwan. More testing is needed to reveal the domestic distribution of these viruses.

The main way the rose viruses spread is likely through vegetative propagation in commercial nurseries and subsequent nationwide distribution to gardens through root grafting. However, some viruses may have other transmission mechanisms once in the environment, including seed, pollen and arthropod vectors [7,55]. ApMV, PNRSV, RoCV-1 and ASGV are reported to be transmitted by seed and pollen [28,47,56,57]. Some ilarviruses can be transmitted by thrips and other flower-visiting arthropods. For instance, PNRSV is spread throughout New Zealand by the western flower thrips (*Frankliniella occidentalis*) [7,47]; however, *F. occidentalis* does not harbor in Taiwan. It cannot be excluded that PNRSV has other insect vectors. The arthropod vector of ApMV, RoCV-1 and ASGV is still unknown. RSDaV is known to be transmitted by rose-grass aphid (*Metopolophium dirhodum*) and yellow rose aphid (*Rhodobium porosum*) [31]. *R. porosum* is one of the common pests of roses in Taiwan [58], which may cause the domestic spread of RSDaV. The transmission mode of RoPV and RIV-2 remains unknown and needs to be investigated. Damage to roses by pests including aphids, mites, thrips and whiteflies could be observed in the surveyed areas. Whether these arthropods are responsible for transmitting the rose viruses needs to be investigated. Although the above-mentioned viruses were not detected in the Changhua and Nantou nurseries, it is still necessary to continue to monitor rose seedlings in commercial nurseries to eliminate the virus source.

To our knowledge, among these rose viruses, ASGV is the only one recorded in Taiwan from pear (*Pyrus pyrifolia* var. *Hengshen*) [59]. This is the first finding of ASGV in Taiwan roses. In conclusion, this is the first report of viruses infecting roses in Taiwan, including ApMV, PNRSV, RIV-2, RoPV, RoCV-1, ASGV and RSDaV, of which RIV-2 is a newly identified species of the genus *Ilarvirus*. The synergy between these viruses was observed in co-infections of RIV-2 with ApMV, ASGV, RoPV and RSDaV, causing severe damage to roses in Chiayi. The rose varieties cultivated in Taiwan are mainly imported from abroad, such as the Netherlands, the USA and Japan, and there is a risk of invasion by foreign pathogenic microorganisms and pests. Although the HTS technology has been a powerful diagnostic tool, the development of detection techniques, such as RT-PCR and ELISA, is also critical for virus inspection. The RT-PCR methods developed in this study can be adapted to detect rose viruses in samples from nurseries, fields and imported materials. Nepoviruses such as ArMV and SLRSV, which are commonly associated with RMD, were not detected in our HTS analysis. These viruses also need to be tested in the future.

## Figures and Tables

**Figure 1 viruses-14-02537-f001:**
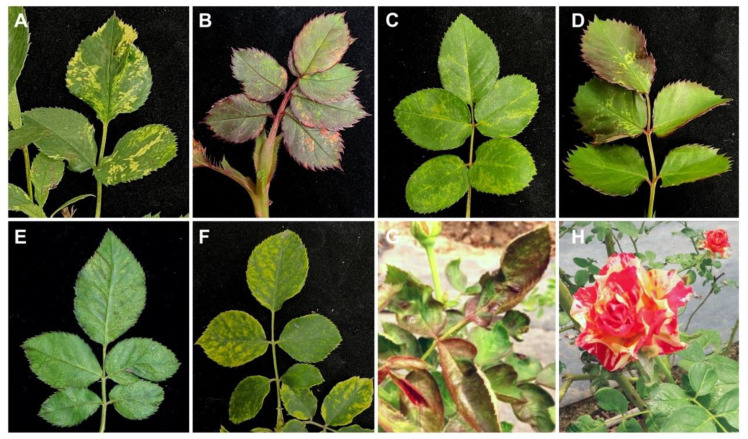
Symptom observations of roses. (**A**) yellow patterns, (**B**) reddish spots, (**C**) mottling with chlorotic patterns, (**D**) yellow spots, (**E**) ringspots, (**F**) mosaic with yellow spots, (**G**) wrinkle-curling on leaves and (**H**) color-breaking on flowers are shown. Viruses detected in the rose samples are listed in Appendix A.

**Figure 2 viruses-14-02537-f002:**
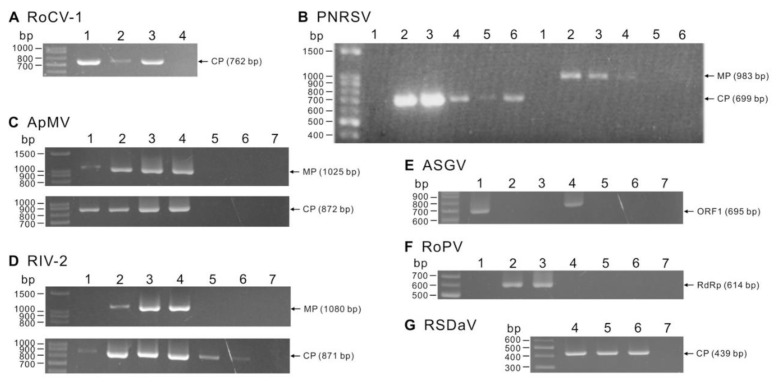
Detection of rose viruses by reverse transcription-polymerase chain reaction. (**A**) rose cryptic virus 1 (RoCV-1), (**B**) prunus necrotic ringspot virus (PNRSV), (**C**) apple mosaic virus (ApMV), (**D**) rose ilarvirus 2 (RIV-2), (**E**) apple stem grooving virus (ASGV), (**F**) rose partitivirus (RoPV), and (**G**) rose spring dwarf-associated virus (RSDaV) were detected in field-collected rose samples. Primers used for virus detection are listed in Appendix A. Numbers represent sample code. The same samples were used to detect ApMV, RIV-2, ASGV, RoPV and RSDaV. Sample #4 in (**A**), sample #1 in (**B**), and sample #7 in (**C**–**G**) are asymptomatic rose leaves used as negative controls. The target genes and expected amplicon sizes are shown and indicated by arrows.

**Figure 3 viruses-14-02537-f003:**
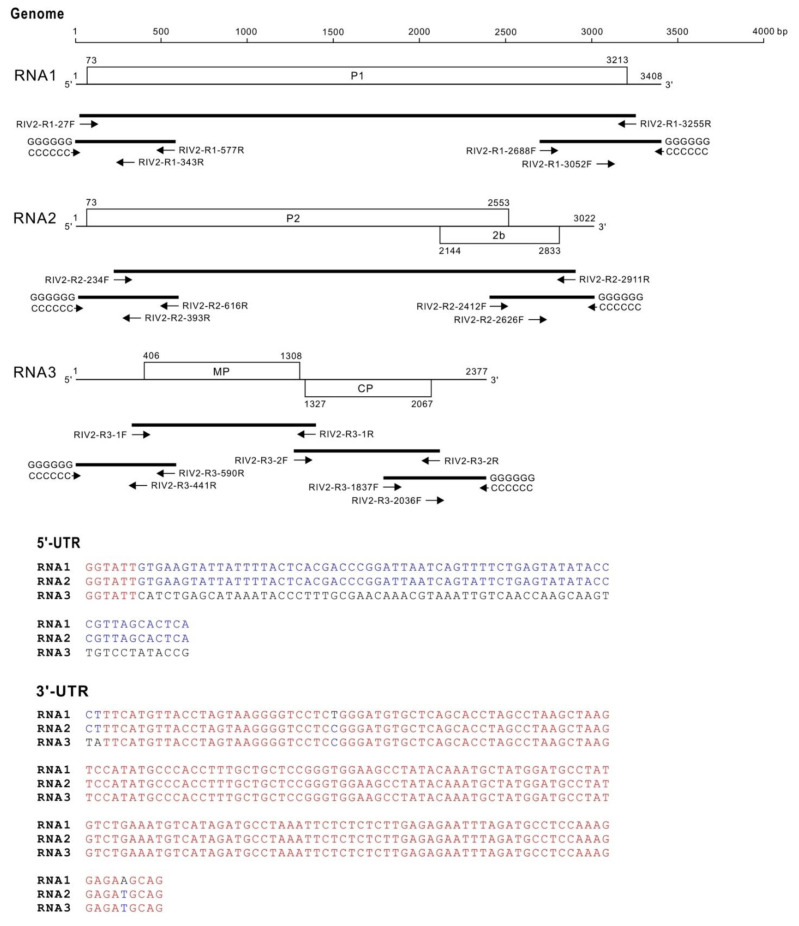
Schematic representation of genome organization of rose ilarvirus 2 (RIV-2). White boxes represent open reading frames (ORFs). Lines represent untranslatable regions (UTRs). The nucleotide positions of the individual ORFs are indicated. Bold lines represent DNA fragments amplified by reverse transcription-polymerase chain reaction (RT-PCR). Primers used for RT-PCR amplification are indicated by arrows. The nucleotide sequences of the primers are shown in Appendix A. Nucleotide sequences of the 5′-untranslatable region (UTR) and 3′-UTR are shown below. Identical nucleotides in the three genomic RNA segments are indicated in red. Nucleotides that are identical in two of the three genomic RNA segments are indicated in blue.

**Figure 4 viruses-14-02537-f004:**
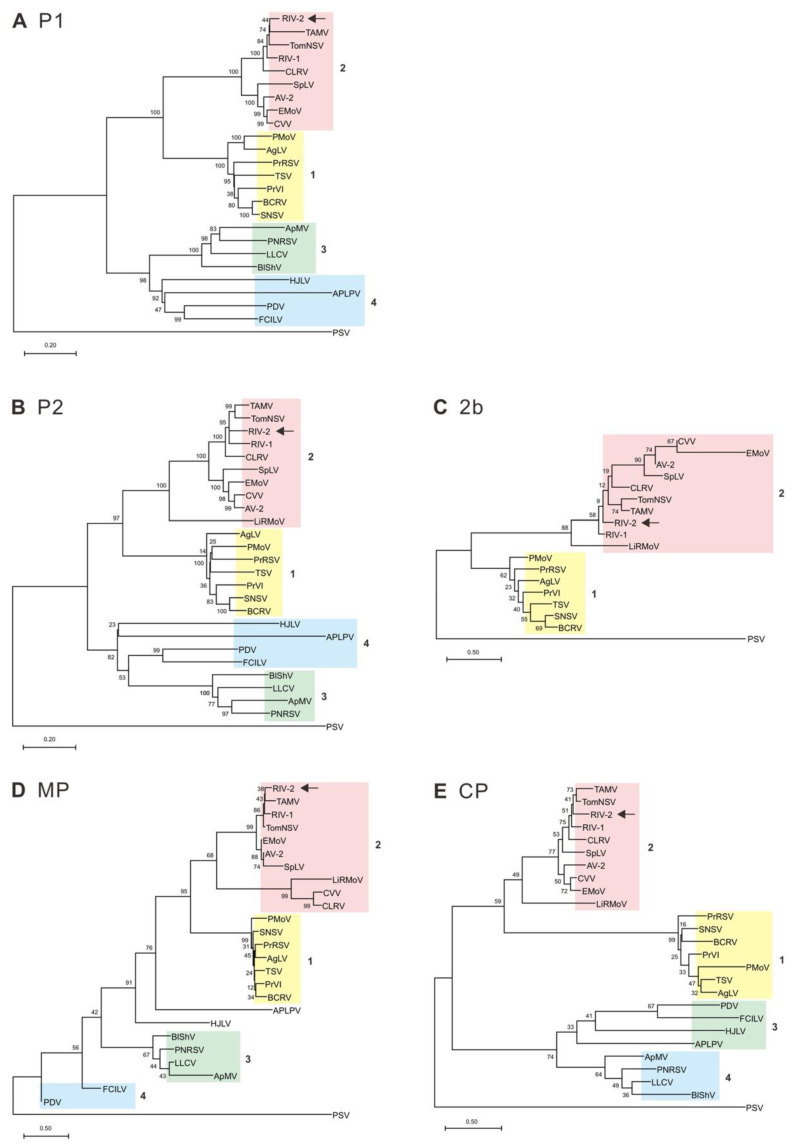
Phylogenetic relationship of the amino acid sequences of (**A**) P1 protein, (**B**) P2 protein, (**C**) 2b protein, (**D**) movement protein (MP) and (**E**) coat protein (CP) of rose ilarvirus 2 (RIV-2) and other ilarviruses. The dendrograms were produced using the Neighbor-Joining algorithm with 1000 bootstrap replicates. Scale refers to amino acid substitutions per site. PSV was used as an outgroup. The newly characterized RIV-2 in this study is indicated by arrows. Subgroups assigned based on additional 2b proteins and phylogenetic relationships are represented by numbers. See Appendix A for virus information.

**Table 1 viruses-14-02537-t001:** The annotation result of the contigs de novo assembled from high-throughput sequencing reads.

Annotated Virus Name	Abbrev.	Taxonomy	Contig No.	Contig Length	Total Reads Count	Nt Identity (%)
Tomato necrotic streak virus	TomNSV	*Ilarvirus*/*Bromoviridae*	4	666–2810	36,412–311,178	79.7–88.0
Apple mosaic virus	ApMV	*Ilarvirus*/*Bromoviridae*	6	570–2027	8149–136,164	90.6–98.4
Prunus necrotic ringspot virus	PNRSV	*Ilarvirus*/*Bromoviridae*	6	586–2578	1228–69,272	96.0–99.3
Rose cryptic virus 1	RoCV-1	Unclassified/*Partitiviridae*	3	1366–1552	2402–16,140	99.9–100
Rose partitivirus	RoPV	Unclassified/*Partitiviridae*	2	1793–1913	2248–2513	98.6–99.4
Rose spring dwarf-associated virus	RSDaV	*Luteovirus*/*Tombusviridae*	2	874–4873	270–2213	89.6–95.5
Rose leaf rosette-associated virus	RLRaV	*Closterovirus*/*Closteroviridae*	20	519–5727	20–1912	77.5–98.4
Apple stem grooving virus	ASGV	*Capillovirus*/*Betaflexiviridae*	3	716–5482	2805–5461	90.6–94.0
Rose virus B	RVB	*Carlavirus*/*Betaflexiviridae*	3	597–831	46–67	92.2–95.5
Plasmopara viticola lesion associated ourmia-like virus 33	PvLaOV-33	*Ourmiavirus*	1	572	20	83.8

**Table 2 viruses-14-02537-t002:** Comparison of nucleotide (nt) and amino acid (aa) sequences of the whole genome sequence of rose ilarvirus 2 (RIV-2) with other *Ilarvirus* species.

Virus ^a^	RNA1	P1		RNA2	P2		2b		RNA3	MP		CP	
	nt%	nt%	aa%	nt%	nt%	aa%	nt%	aa%	nt%	nt%	aa%	nt%	aa%
RIV-2	100	100	100	100	100	100	100	100	100	100	100	100	100
AgLV	55.2	55.9	43.3	53.1	51.6	38.5	46.2	19.1	44.0	48.1	26.9	43.2	19.4
APLPV	51.1	52.0	31.2	40.4	43.0	27.8	- ^b^	-	42.3	46.4	14.0	46.4	18.6
ApMV	49.2	51.2	35.0	41.3	41.8	28.8	-	-	44.0	45.0	17.1	42.0	15.5
AV-2	70.1	67.2	72.2	66.8	64.2	63.2	49.5	41.5	64.1	65.8	68.8	55.3	51.0
BCRV	54.6	55.2	45.0	50.2	50.4	38.5	46.1	16.1	45.1	47.5	25.9	48.4	18.7
BlShV	51.3	51.9	35.2	41.5	45.6	30.5	-	-	40.7	44.2	18.1	47.3	16.1
CLRV	78.4	77.8	85.3	75.8	73.7	75.1	73.4	69.6	62.6	48.3	27.9	57.4	59.2
CVV	69.4	66.7	72.3	67.4	65.4	71.8	50.1	37.0	59.9	49.8	30.2	56.1	52.6
EMoV	69.2	67.8	72.4	67.0	64.3	63.5	48.1	34.3	65.4	66.0	69.1	56.2	49.4
FCILV	51.4	51.6	35.4	38.2	41.3	31.4	-	-	45.9	45.2	20.5	40.3	16.9
HJLV	50.4	50.7	33.0	37.1	39.0	27.7	-	-	39.8	45.8	18.9	45.1	15.1
LiRMoV	-	-	-	57.3	58.0	50.8	48.1	32.3	50.9	46.5	27.7	45.2	29.0
LLCV	51.8	52.4	35.2	37.1	40.4	29.0	-	-	44.5	46.6	18.5	45.3	16.3
PDV	51.9	52.7	33.6	41.1	42.8	33.5	-	-	45.6	44.3	27.1	42.8	15.3
PMoV	53.5	54.3	43.6	51.3	49.6	37.9	44.3	17.0	44.2	47.1	26.9	41.4	16.6
PNRV	51.2	52.3	35.4	40.0	43.6	29.6	-	-	41.4	45.5	17.5	42.1	15.2
PrRSV	53.4	55.0	42.9	50.6	49.6	38.5	45.8	19.1	42.3	44.8	23.6	44.6	16.5
PrVI	54.5	56.0	44.6	51.4	50.1	39.0	45.5	19.6	46.4	47.9	25.0	42.9	17.7
RIV-1	79.4	79.7	89.2	76.4	74.3	75.3	71.3	70.7	76.8	76.8	77.5	63.4	65.0
SNSV	54.3	55.5	45.3	51.3	51.2	38.3	46.8	17.8	47.4	47.4	23.9	44.7	17.2
SpLV	69.5	68.2	70.2	66.1	63.8	60.9	48.6	39.3	65.3	63.5	59.1	55.8	50.8
TAMV	80.2	80.1	82.0	74.3	71.4	71.3	63.8	50.4	72.3	74.1	75.7	64.3	63.6
TomNSV	81.9	84.3	87.1	77.3	80.2	83.4	78.1	74.2	79.8	79.8	81.4	78.6	80.7
TSV	53.7	54.6	43.1	52.0	51.1	40.4	46.6	17.3	45.4	48.8	24.3	47.1	17.8

^a^ See Appendix A for virus names. ^b^ “-” represents the sequence is unavailable in the GenBank database.

**Table 3 viruses-14-02537-t003:** Field survey results of rose viruses in central Taiwan.

Region	Collection Date	Sample No.	Virus-Positive No.						
			ApMV	PNRSV	RIV-2	RoPV	RoCV-1	ASGV	RSDaV
Chiayi (Organic farm)	2 June 2020	5	0	5	0	0	0	0	0
	17 September 2020	15	0	2	0	0	0	0	0
	2 February 2021	12	4	9	6	2	0	2	3
	29 July 2021	29	1	0	4	1	2	0	1
	21 January 2022	12	1	0	1	0	1	0	2
	26 August 2022	9	0	0	0	0	0	0	0
Total no. (Detection rate)		82	6 (7.3%)	16 (19.5%)	11 (13.4%)	3 (3.7%)	3 (3.7%)	2 (2.4%)	6 (7.3%)
Taichung (AU campus)	12 July 2021	5	0	0	0	2	2	0	0
	22 September 2022	5	0	0	0	0	2	0	1
Total no. (Detection rate)		10	0	0	0	2 (20.0%)	4 (40.0%)	0	1 (10.0%)
Sum (Detection rate)		92	6 (6.5%)	16 (17.4%)	11 (12.0%)	5 (5.4%)	7 (7.6%)	2 (2.2%)	7 (7.6%)

## Data Availability

The genomic sequences of RIV-2 have been lodged in GenBank.

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
