# Peer review of "Rose Virome Analysis and Identification of a Novel Ilarvirus in Taiwan"

_viruses, 2022, doi:10.3390/v14112537_

Round 1
Reviewer 1 Report
This is an excellent review on rose viruses in Taiwan. The authors did a thorough survey of the occurrence of different rose viruses in an organic farm that produce roses in Central Taiwan Chaiyi. Through RNA-seq and reverse transcription-polymerase chain reaction validation, a total of seven viruses were detected, including six known rose viruses, namely apple mosaic virus, prunus necrotic ringspot virus, rose partitivirus, apple stem grooving virus, rose spring dwarf-associated virus and rose cryptic virus 1, and a novel ilarvirus.tentatively named rose ilarvirus 2. It is the first report of the rose viruses in Taiwan.
Author Response
The reviewer has no comment.
Reviewer 2 Report
The manuscript describes the identification of viruses by High Through Sequencing of total RNA and its confirmation by RT-PCR in 12 samples of roses (R. hybrida) with viral symptoms. Samples were collected from a rose organic farm from Fanlu Township, Chiayi County, and the campus of Asia University in Wufeng District, Taichung City, Taiwan. Seven viruses were identified, six known viruses, and one putative new species. This later was molecularly characterized and evidence indicates that corresponds to a putative new species of subgroup 2 of the genus Ilarvirus, which was tentatively named rose Ilarvirus 2. The identified viruses were subsequently tested and detected by RT-PCR on 92 samples. The contribution has scientific relevance. However, it requires some precision in the methodology and contains some inaccuracies in the way the new species of Ilarvirus is named in the different sections, tables, and figures of the contribution. Some specific comments are included below. These and other comments and suggestions were directly included in the manuscript.

Reviewer 3 Report
In this work, the authors describe rose virome and identification of a novel ilarvirus in Taiwan. The experiments and results obtained are interesting and worth publishing. Overall the study is well designed and performed, report is well written, and I appreciate the use of RACE, individual RT-PCR, and sequencing to confirm the presence of various viruses and construction of the whole genome for the new ilarvirus. As ICT has not laid down any specific level of sequence similarity it may be considered a new member.
However there are a few minor points that are mention in attached file may be updated by the authors. Discussion should focus on the analysis and highlight of the result.
